# The Effectiveness of Face-to-Face versus Online Delivery of Continuing Professional Development for Science Teachers: A Systematic Review

**Zhi Li [1,2,*], Norlizah Che Hassan [1] and Habibah Ab. Jalil [1]**

1   Faculty of Educational Studies, Universiti Putra Malaysia, Serdang 43400, Malaysia
2   School of Media and Art Design, Guilin University of Aerospace Technology, Guilin 541004, China
*   Correspondence: gs64815@student.upm.edu.my

**Abstract:** Purpose: This systematic review assesses the effectiveness of face-to-face and online delivery modes of continuing professional development (CPD) for science teachers. It focuses on three aspects: evaluating the effectiveness of these modes, summarizing the literature on the factors influencing them, and conducting a comparative analysis of their advantages. Methods: The research team employed the Preferred Reporting Items for Systematic Reviews and Meta-Analyses (PRISMA) method and the Mixed Methods Appraisal Tool (MMAT) for article quality assessment. A total of 12 articles, selected from a potential 82, were included in the study. Results: This research suggests that the face-to-face and online CPD modes are equally effective and that external factors, such as psychological variables and establishing communication communities, influence their effectiveness. Face-to-face CPD fosters communication communities, while online CPD offers geographical flexibility and cost-saving benefits. Implications: The effectiveness of face-to-face and online CPD relies on external psychosocial factors. Future research should focus on strategies to enhance participants' communication engagement in online communities. Additionally, it is worth conducting further investigations of the potential relationships between psychosocial variables and the effectiveness of online CPD, along with the impact of digital skills on online CPD.

**Keywords:** continuing professional development (CPD); effectiveness; face-to-face; online delivery; science teachers

## 1. Introduction

Continuing professional development (CPD) refers to various learning activities to enhance professional capabilities [1,2]. Globally, governments and schools use CPD to improve teachers' capabilities and the quality of teaching, thereby effectively promoting educational quality and reaching a consensus. CPD proves to be an effective means of elevating teachers' pedagogical skills and instructional competence. For educators, professional development is widely recognized as a planned, continuous, and lifelong learning process upon which educational quality is enhanced [3]. During this process, teachers' personal qualities and real-world practice are improved [4]. Teachers should be lifelong learners who must adapt and proficiently embrace new teaching methodologies and emerging tools over time. This adaptation is essential to ensuring their ability to seamlessly integrate new technology into the classroom with the highest level of quality [5,6].

The field of CPD research has yielded notable academic achievements [7,8]. Through CPD training, educators not only enrich their knowledge in their respective fields but also cultivate the improvement of teaching practices [9]. The significance and value of CPD for guaranteeing high-quality education has garnered substantial attention from researchers [10], with many studies on CPD being conducted in the educational field. The research designs of such studies exhibit a wide scope; for example, those that evaluate the impact of CPD projects on teaching practice and offer analyses of the factors that hinder

effective participation. Additionally, research on CPD explores enhancements and policy frameworks in an attempt to solve the shortcomings of CPD initiatives [11–13].

The development of computer and networking technology has triggered some highly significant social changes, and the field of education has changed. The combination of education and technology has always been a focus of research; researchers have also proposed changes to the delivery modes of education involving the use of computer and networking technology, such as E-learning [14]. Harnessing the power of computers and network connections, learning resources can be easily accessed from almost any location [15]. E-learning has also led to new insights into the evolution of CPD-related fields [16,17]. The COVID-19 pandemic forced governments and educational institutions to adapt quickly, and the modes of E-learning, based on the Internet, assumed the role of maintaining the stable operation of the education system. The effective implementation of online education during the pandemic has generated new ideas for delivering CPD over the Internet, and research results have confirmed the success of the online CPD delivery mode [18–21]. In the post-COVID-19 era, there has been a notable surge in the initiation of CPD projects across diverse fields through online platforms. This approach not only adeptly mitigates the constraints inherent in face-to-face CPD, such as time schedule and geographical limitations [22–25]. This raises a fundamental question: are there indispensable aspects of face-to-face CPD delivery modes? This has sparked a series of reflections. For instance, can online CPD effectively enhance teachers' professional knowledge and practical skills in comparison to traditional modes? What factors influence these two delivery methods, and are they consistent? Can online teaching models be leveraged for the professional development of educators in the future? These questions constitute the foundation of the present study. To address them, the research team conducted a systematic literature review that comprehensively assessed and analyzed the existing research findings regarding CPD across different delivery modes.

There are many research papers on CPD in the field of education. To ensure that this article's focus is unambiguous, the research team focused on studies relevant to science teachers; this encompassed various disciplines, including science, technology, engineering, mathematics, biology, and chemistry. This decision was based on two considerations. First, science education is the cornerstone of a country's competitiveness, especially during the K-12 education stage, where science education is of paramount importance [26]. Second, many countries, such as the United States, China, and India, have a shortage of high-quality teachers in science fields due to geographical territories and economic development levels. This shortage is more serious in high-poverty and rural areas; teachers seeking employment in these areas are generally less able to easily complete the high-quality requirements for all subjects they teach and are also unable to access content-specific professional development [27].

Effective CPD has the potential to change teachers' practices and enhance students' learning outcomes [28], but determining its effectiveness can be challenging because it is subjective; what works for one participant may not work for another. However, evaluating the efficacy of CPD is crucial not only for personal and professional development but also for determining whether CPD activities should be revisited. To this end, researchers have attempted to create various evaluation frameworks for assessing the effectiveness of CPD.

In 1975, Donald Kirkpatrick introduced an evaluation model in his book *Evaluating Training Programs*, comprising four core levels: reaction, learning, behavior, and results [29]. Kirkpatrick's model has since garnered significant recognition as a tool for gauging the effectiveness of training programs, including teachers' CPD; however, it does have certain limitations. First, the model places significant emphasis on quantitative measurements, yet the sections on behavior and results can be somewhat abstract and challenging to quantify. Second, it lacks an assessment of social variables and the educational organizational climate. The model is general and not specific to the educational field.

In 2000, Thomas Guskey provided a method primarily focused on specifically evaluating CPD's effectiveness in the educational field. This method comprises five levels,

providing a strong basis for ensuring that a CPD offering is effective. These include (1) participants' reactions, (2) participants' learning, (3) organization and support, (4) participants' use of new knowledge or skills, and (5) student learning outcomes [30]. Additionally, Guskey summarized thirteen characteristics of effective CPD, as follows: (1) providing sufficient time and resources; (2) promoting collegiality and collaboration; (3) including procedures for evaluation; (4) modeling high-quality instruction; (5) being based in schools or on site; (6) building leadership capacity; (7) being built on the identified needs of the teachers; (8) being driven by analyses of student learning data; (9) focusing on individual and organizational improvement; (10) including follow-up and support; (11) being ongoing and embedded in the job; (12) taking a variety of forms; and (13) promoting continuous inquiry and reflection [31]. Guskey's approach exhibits limitations, particularly in its neglect of contextual factors such as student characteristics, teacher attributes, and school features. Desimone's suggested evaluative framework mitigates this constraint.

Desimone [32] provided a comprehensive framework for evaluating the effects of professional development. He thought that there were three aspects of professional development relevant to the measurement of its effectiveness: (1) the core features of effective professional development are content-focused active learning, coherence, duration, and collective participation; (2) the way this effective professional development affects teachers' knowledge, their practice, and students' learning; and (3) contextual factors, such as student characteristics, teacher characteristics, and school characteristics, are related to the effectiveness of professional development [32,33]. This conceptual framework encompasses the core elements of effective professional development while also considering the factors that mediate and moderate its impact. These factors include environment variables and the characteristics of the teachers themselves.

Darling-Hammond and McLaughlin [34] provided a set of essential characteristics necessary for optimal professional development. These attributes comprise (1) engaging teachers in the concrete tasks of teaching, assessment, observation, and reflection that illuminate the processes of learning and development; (2) being grounded in inquiry, reflection, and participant-driven experimentation; (3) being collaborative, involving a sharing of knowledge among educators and a focus on teachers' communities of practice rather than on individual teachers; (4) being connected to and derived from teachers' work with their students; (5) being sustained, ongoing, intensive, and supported by modeling, coaching, and the collective solving of specific problems in practice; and (6) being connected to other aspects of school change [34].

In 2017, Ravitz and his co-authors outlined seven prevailing characteristics that delineate successful teacher professional development programs: (1) a primary focus on the content; (2) the application of active learning strategies grounded in adult learning theory; (3) the encouragement of collaboration, often embedded within teachers' work settings; (4) the utilization of models and demonstrations to illustrate effective teaching practices; (5) the provision of coaching and expert guidance; (6) the establishment of opportunities for feedback and self-reflection; and (7) a sustained duration [35].

From the work outlined above, it can be concluded that there are two main avenues for evaluating the effectiveness of CPD. One approach focuses on assessing the impact of CPD based on participants' performance by measuring student learning outcomes. The second approach centers around evaluating the effectiveness of CPD through collecting teachers' personal feedback and reflection reports. The distinct pathways of gauging CPD effectiveness, ranging from objective measurements of student learning outcomes to educators' subjective reflections, underscore this educational phenomenon's complexity.

### 1.1. Definition of Terms

In the context of this discussion, certain key terms are defined to provide clarity and understanding. CPD can be delivered in two types: face-to-face or via the Internet and a combination of a 'Hybrid' [36].

### 1.1.1. Face-to-Face Delivery of CPD

Face-to-face mode is a traditional way of delivering CPD. This mode can take various forms, such as workshops, seminars, and in-service training [37]. In this form of CPD, the project needs to be carried out at the same location at a specified time [38].

### 1.1.2. Online Delivery of CPD

Unlike the face-to-face delivery of CPD, participants can engage in online CPD activities based on their schedule and location as long as they can access the Internet using a computer [39,40]. The delivery modes of online CPD are more diverse, such as online meetings, Massive Open Online Courses (MOOCs), blended learning, etc. [17,41].

## 2. Research Questions

This study aims to provide a comprehensive analysis of the effectiveness of both online CPD and face-to-face CPD, as well as their impact on science educators, by conducting a systematic review. It aims to further explore issues in the field of online CPD and provide valuable insights, with a particular focus on the following questions:

RQ1. How does the effectiveness of face-to-face CPD compare to that of online CPD for science educators?
RQ2. What factors could potentially impact the efficacy of diverse forms of CPD programs?
RQ3. What are the advantages of different CPD delivery modes?

## 3. Methodology

This study employed the systematic literature review (SLR) method to ensure that future scholars can replicate this research and conduct further studies. Throughout our systematic literature review, we adhered to the relevant guidelines provided by the Preferred Reporting Items for Systematic Reviews and Meta-Analyses (PRISMA) [42,43]. The research team conducted a systematic research process to address the questions posed in this study, including literature retrieval, organization, and analysis.

### 3.1. Inclusion and Exclusion Criteria

Inclusion and exclusion criteria were established to clearly compare the effectiveness of both traditional and online CPD models for science teachers and investigate the impact of various CPD delivery modes on the effectiveness of science teachers. The specific criteria for inclusion and exclusion are detailed in Table 1.

**Table 1.** Inclusion and exclusion criteria.

| Inclusion Criteria | Exclusion Criteria |
| --- | --- |
| (1) English-language research published in peer-reviewed journals, including both research articles and conference papers;<br>(2) Article focuses on the effectiveness of different delivery types of CPD in improving science teacher quality and practice;<br>(3) Concerns science teachers or science educators. | (1) Book reviews, reports, and degree dissertations;<br><br>(2) Studies involving subjects other than science teachers, such as physical education instructors. |

### 3.2. Data Source

Electronic databases, specifically Web of Science (WoS), Scopus, ERIC, and ScienceDirect, were searched for the relevant literature published up to 12 October 2023 (these articles are included and updated in the database).

The research team chose these databases because they are internationally renowned academic literature databases that feature numerous peer-reviewed journals, indicating that the selected articles have undergone expert review, are of high quality, and their research findings can be relied upon. Additionally, these databases are strongly aligned with the research field, particularly the field of education, offering extensive coverage and rich

resources; researchers can thus easily access documents related to their research topics. They also have global coverage and encompass research outcomes from various fields and regions, facilitating the acquisition of diverse and authoritative information on a global scale. Hence, selecting these four databases ensures the reliability and depth of the research.

### 3.3. Search Strategy

The primary objective of this study was to investigate and compare the impact of face-to-face and online CPD on science teachers' literacy and practical capabilities. Recognizing the varied terminology used in fields closely related to CPD, the research team employed a comprehensive search strategy that integrated all pertinent search terms into the search algorithm. In addition, we diligently identified synonyms and commonly used alternatives to the term "CPD". This meticulous approach resulted in the refined search terms presented in Table 2.

**Table 2.** Refined search terms.

| Sections | Terms |
|---|---|
| Keywords related to the topic | Continuing professional development, continuous professional development, CPD |
| Study population | Science teachers |
| Type of CPD delivery | Face-to-face, traditional, online |
| Terms related to effectiveness | Effectiveness, efficacy |

These keywords were combined using the Boolean operators AND and OR. The search strings used for database retrieval are summarized and presented below:

*("Continuing Professional Development" OR "Continuous Professional Development" OR "CPD")*

*AND*

*Science Teachers*

*AND*

*("Face-to-face" OR "Online")*

*AND*

*("Effectiveness" OR "Efficacy")*

This comprehensive approach allowed us to retrieve the relevant literature, forming the foundation for our systematic review. Each research team member applied the search strings for literature retrieval in the databases, with summary judgments made against each of the above criteria. The team then met to review all the decisions depicted in Table 3.

**Table 3.** Search strategy and number of studies (results returned).

| Database | In | Publication Data | N |
|---|---|---|---|
| Web of Science (WoS) | Title, abstract, and keywords | Not specified in the search | 2 |
| Scopus | Title, abstract, and keywords | Not specified in the search | 2 |
| ERIC | Title, abstract, and keywords | 2004–2023 | 26 |
| ScienceDirect | Title, abstract, and keywords | Not specified in the search | 52 |

### 3.4. Data Extraction

During the data extraction stage, the research team used EndNote20 software to edit and organize the documents and Excel software (version 2022) to summarize and manage the results.

After conducting the systematic review, the preliminary reviewed articles were imported into Endnote20 for retention, and 82 manuscripts were retained; of these, 23 articles did not meet the first inclusion criterion related to the article type criterion, and 2 were duplicate manuscripts. Those documents were excluded. Then, the research team reviewed the articles' titles, abstracts, and keywords and deleted 29 articles. These articles failed to meet the criterion, as these articles did not align with the research focus specified in

the second inclusion criterion. Next, the team members further reviewed the full text of the remaining articles, 16 of which did not meet the second inclusion criteria and so were excluded. A quality assessment was performed on the remaining 12 documents. Figure 1 shows a PRISMA flowchart reporting the number of articles included and excluded at each step.

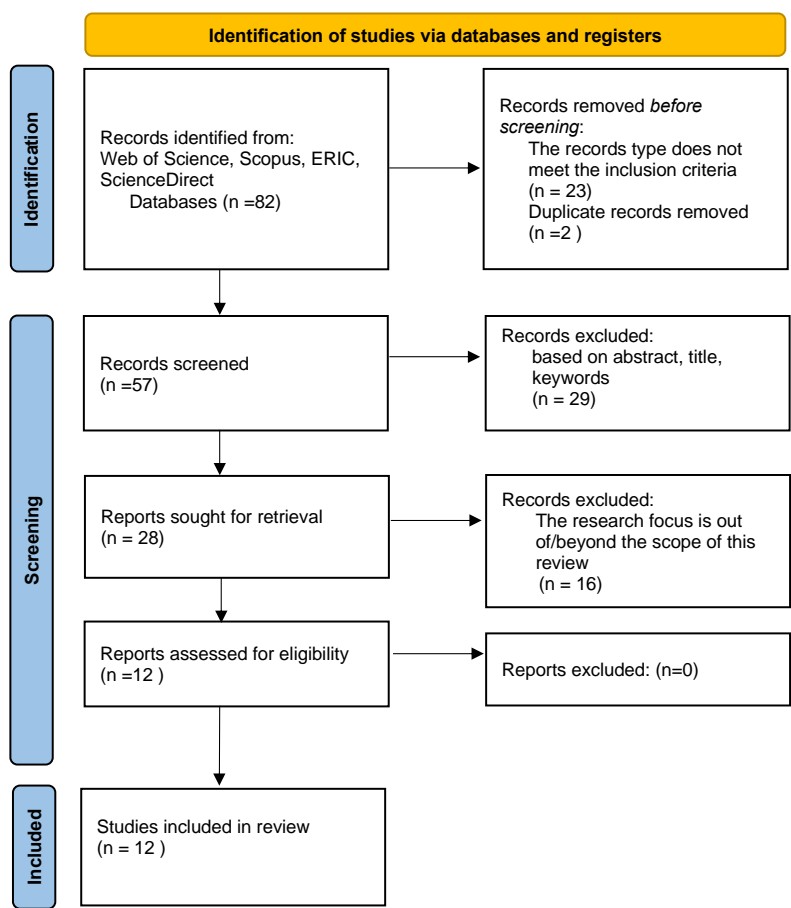

**Figure 1.** Flowchart of the systematic review selection process based on the PRISMA flow diagram.

*3.5. Quality Assessment*

The Mixed Methods Appraisal Tool (MMAT) was used to judge the quality of the studies. The MMAT is an important tool for assessing the quality of studies based on five evaluation criteria and is used during the assessment stage of systematic reviews, that is, reviews containing studies using mixed methodologies and quantitative and qualitative approaches [44,45].

Quality assessment involves two steps: First, the research team screens the articles, categorizes them based on five categories of study design (e.g., mixed methods, qualitative, etc.), and then evaluates them based on the five quality criteria associated with each category. In addition, the research team should determine whether these documents should be included in the final review assessment. At this juncture, documents must satisfy both conditions to progress to the final evaluation phase. Per the MMAT User Guide, papers must meet two screening questions outlined by MMAT before undergoing the conclusive assessment. Responding with "No" or "Can't tell" to either or both questions could signify that the paper is unsuitable for appraisal using the MMAT. Please refer to Table 4 for further details. Second, each category is evaluated based on five criteria associated with it. Each criterion can be rated in three ways: Y for yes, N for no, and CT for cannot tell. During the second-stage screening, when the answer is "no" or "cannot tell" to one or both screening questions, the steps detailed in Table 5 are followed.

**Table 4.** Based on the first MMAT screening stage.

| No. | Category of Study Designs | Study | Methodological Quality Criteria | YES | NO | CANNOT TELL | Comments |
|---|---|---|---|---|---|---|---|
| 1 | Quantitative descriptive studies | (Beardsley et al., 2022) [19] | S1. Are there clear research questions? | ✓ | | | |
| | | | S2. Do the collected data allow us to address the research questions? | ✓ | | | |
| 2 | Mixed-methods studies | (Binmohsen and Abrahams, 2020) [21] | S1. Are there clear research questions? | ✓ | | | |
| | | | S2. Do the collected data allow us to address the research questions? | ✓ | | | |
| 3 | Mixed-methods studies | (Bitan-Friedlander et al., 2004) [46] | S1. Are there clear research questions? | ✓ | | | |
| | | | S2. Do the collected data allow us to address the research questions? | ✓ | | | |
| 4 | Qualitative studies | (Haydn and Barton, 2008) [47] | S1. Are there clear research questions? | ✓ | | | |
| | | | S2. Do the collected data allow us to address the research questions? | ✓ | | | |
| 5 | Qualitative studies | (Arce et al., 2014) [48] | S1. Are there clear research questions? | ✓ | | | |
| | | | S2. Do the collected data allow us to address the research questions? | ✓ | | | |
| 6 | Qualitative studies | (Juuti et al., 2023) [49] | S1. Are there clear research questions? | ✓ | | | |
| | | | S2. Do the collected data allow us to address the research questions? | ✓ | | | |
| 7 | Quantitative non-randomized studies | (Mary and Cha, 2021) [50] | S1. Are there clear research questions? | ✓ | | | |
| | | | S2. Do the collected data allow us to address the research questions? | ✓ | | | |
| 8 | Mixed-methods studies | (Stevenson et al., 2015) [27] | S1. Are there clear research questions? | ✓ | | | |
| | | | S2. Do the collected data allow us to address the research questions? | ✓ | | | |
| 9 | Qualitative studies | (Owston et al., 2008) [51] | S1. Are there clear research questions? | ✓ | | | |
| | | | S2. Do the collected data allow us to address the research questions? | ✓ | | | |
| 10 | Mixed-methods studies | (Lichtenstein and Phillips, 2021) [18] | S1. Are there clear research questions? | ✓ | | | |
| | | | S2. Do the collected data allow us to address the research questions? | ✓ | | | |
| 11 | Quantitative descriptive studies | (Ravitz et al., 2017) [35] | S1. Are there clear research questions? | ✓ | | | |
| | | | S2. Do the collected data allow us to address the research questions? | ✓ | | | |
| 12 | Qualitative studies | (Herbert et al., 2016) [52] | S1. Are there clear research questions? | ✓ | | | |
| | | | S2. Do the collected data allow us to address the research questions? | ✓ | | | |

"✓" indicates that the standard is met.

**Table 5.** Based on MMAT's second screening stage.

| Category of Study Designs | Study | Methodological Quality Criteria | YES | NO | CANNOT TELL | Comments |
|---|---|---|---|---|---|---|
| Quantitative descriptive studies | (Beardsley et al., 2022) [19] | Is the sampling strategy relevant to the research question? | ✓ | | | |
| | | Is the sample representative of the target population? | ✓ | | | |
| | | Are the measurements appropriate? | ✓ | | | |
| | | Is the risk of nonresponse bias low? | ✓ | | | |
| | | Is the statistical analysis appropriate to the research question? | ✓ | | | |

**Table 5.** *Cont.*

| Category of Study Designs | Study | Methodological Quality Criteria | Responses | | | |
|---|---|---|---|---|---|---|
| | | | YES | NO | CANNOT TELL | Comments |
| Quantitative non-randomized studies | (Mary and Cha, 2021) [50] | Are the participants representative of the target population? | ✓ | | | |
| | | Are the measurements appropriate regarding both the outcome and intervention (or exposure)? | ✓ | | | |
| | | Are there complete outcome data? | ✓ | | | |
| | | Are the confounders accounted for in the design and analysis? | ✓ | | | |
| | | During the study period, is the intervention administered (or does exposure occur) as intended? | ✓ | | | |
| Mixed-methods studies | (Binmohsen and Abrahams, 2020) [21] | Is there an adequate rationale for using a mixed methods design to address the research question? | ✓ | | | |
| | | Are the different components of the study effectively integrated to answer the research question? | ✓ | | | |
| | | Are the outputs of the integration of qualitative and quantitative components adequately interpreted? | ✓ | | | |
| | | Are divergences and inconsistencies between quantitative and qualitative results adequately addressed? | ✓ | | | |
| | | Do the different components of the study adhere to the quality criteria of each tradition of the methods involved? | ✓ | | | |
| | (Bitan-Friedlander et al., 2004) [45] | Is there an adequate rationale for using a mixed methods design to address the research question? | ✓ | | | |
| | | Are the different components of the study effectively integrated to answer the research question? | ✓ | | | |
| | | Are the outputs of the integration of qualitative and quantitative components adequately interpreted? | ✓ | | | |
| | | Are divergences and inconsistencies between quantitative and qualitative results adequately addressed? | ✓ | | | |
| | | Do the different components of the study adhere to the quality criteria of each tradition of the methods involved? | ✓ | | | |
| | (Stevenson et al., 2015) [27] | Is there an adequate rationale for using a mixed methods design to address the research question? | ✓ | | | |
| | | Are the different components of the study effectively integrated to answer the research question? | ✓ | | | |
| | | Are the outputs of the integration of qualitative and quantitative components adequately interpreted? | ✓ | | | |
| | | Are divergences and inconsistencies between quantitative and qualitative results adequately addressed? | ✓ | | | |
| | | Do the different components of the study adhere to the quality criteria of each tradition of the methods involved? | ✓ | | | |
| | (Lichtenstein and Phillips, 2021) [18] | Is there an adequate rationale for using a mixed methods design to address the research question? | ✓ | | | |
| | | Are the different components of the study effectively integrated to answer the research question? | ✓ | | | |
| | | Are the outputs of the integration of qualitative and quantitative components adequately interpreted? | ✓ | | | |
| | | Are divergences and inconsistencies between quantitative and qualitative results adequately addressed? | ✓ | | | |
| | | Do the different components of the study adhere to the quality criteria of each tradition of the methods involved? | ✓ | | | |
| | (Ravitz et al., 2017) [35] | Is there an adequate rationale for using a mixed methods design to address the research question? | ✓ | | | |
| | | Are the different components of the study effectively integrated to answer the research question? | ✓ | | | |
| | | Are the outputs of the integration of qualitative and quantitative components adequately interpreted? | ✓ | | | |
| | | Are divergences and inconsistencies between quantitative and qualitative results adequately addressed? | ✓ | | | |
| | | Do the different components of the study adhere to the quality criteria of each tradition of the methods involved? | ✓ | | | |

**Table 5.** *Cont.*

| Category of Study Designs | Study | Methodological Quality Criteria | Responses | | | |
|---|---|---|---|---|---|---|
| | | | YES | NO | CANNOT TELL | Comments |
| Qualitative studies | (Haydn and Barton, 2008) [46] | Is the qualitative approach appropriate to the research question? | ✓ | | | |
| | | Are the qualitative data collection methods adequate to address the research question? | ✓ | | | |
| | | Are the findings adequately derived from the data? | ✓ | | | |
| | | Is the interpretation of results sufficiently substantiated by the data? | ✓ | | | |
| | | Is there coherence between qualitative data sources, collection, analysis, and interpretation? | ✓ | | | |
| | (Arce et al., 2014) [47] | Is the qualitative approach appropriate to the research question? | ✓ | | | |
| | | Are the qualitative data collection methods adequate to address the research question? | ✓ | | | |
| | | Are the findings adequately derived from the data? | ✓ | | | |
| | | Is the interpretation of results sufficiently substantiated by the data? | ✓ | | | |
| | | Is there coherence between qualitative data sources, collection, analysis, and interpretation? | ✓ | | | |
| | (Juuti et al., 2023) [48] | Is the qualitative approach appropriate to the research question? | ✓ | | | |
| | | Are the qualitative data collection methods adequate to address the research question? | ✓ | | | |
| | | Are the findings adequately derived from the data? | ✓ | | | |
| | | Is the interpretation of results sufficiently substantiated by the data? | ✓ | | | |
| | | Is there coherence between qualitative data sources, collection, analysis, and interpretation? | ✓ | | | |
| | (Owston et al., 2008) [51] | Is the qualitative approach appropriate to the research question? | ✓ | | | |
| | | Are the qualitative data collection methods adequate to address the research question? | ✓ | | | |
| | | Are the findings adequately derived from the data? | ✓ | | | |
| | | Is the interpretation of results sufficiently substantiated by the data? | ✓ | | | |
| | | Is there coherence between qualitative data sources, collection, analysis, and interpretation? | ✓ | | | |
| | (Herbert et al., 2016) [52] | Is the qualitative approach appropriate to the research question? | ✓ | | | |
| | | Are the qualitative data collection methods adequate to address the research question? | ✓ | | | |
| | | Are the findings adequately derived from the data? | ✓ | | | |
| | | Is the interpretation of results sufficiently substantiated by the data? | ✓ | | | |
| | | Is there coherence between qualitative data sources, collection, analysis, and interpretation? | ✓ | | | |

"✓" indicates that the standard is met.

### 4. Results

According to this review, 12 studies referred to the effectiveness of different delivery modes of CPD for science teachers. The research design categories in Table 6 have been adjusted to correspond with those presented in the MMAT. Hong, Pluye, et al. [44] classified qualitative data collection and analysis methods that include case studies, such as focus groups, in-depth interviews, and hybrid thematic analyses (deductive and inductive), into Category 1 (qualitative studies). Studying the association between health-related outcomes and other factors at a specific point in time using cross-sectional analytical methods falls under Category 3 (quantitative non-randomized research). A "Survey Research method by which information is gathered by asking people questions on a specific topic and the data collection procedure is standardized and well defined" [44] belongs to Category 4 (quantitative descriptive studies). Research that involves combining qualitative (QUAL) and quantitative (QUAN) methods belongs to Category 5 (mixed-methods studies). The effectiveness of delivery modes of CPD is usually investigated using qualitative and mixed research approaches.

**Table 6.** Research design of the included studies.

| MAAT Categories of Research Design | N |
| --- | --- |
| Quantitative descriptive studies | 1 |
| Quantitative non-randomized studies | 1 |
| Qualitative studies | 5 |
| Mixed methods studies | 5 |

As shown in Figure 2, most of the studies included in this review were conducted in the United States of America; there were four research articles. Represented by only one research article each, the countries featured in the study included the United Kingdom, Saudi Arabia, South Korea, Puerto Rico, Israel, Finland, Canada, and Australia. It is evident that scholarly attention in America towards studying the effectiveness of various CPD modes for science teachers surpasses that of other nations.

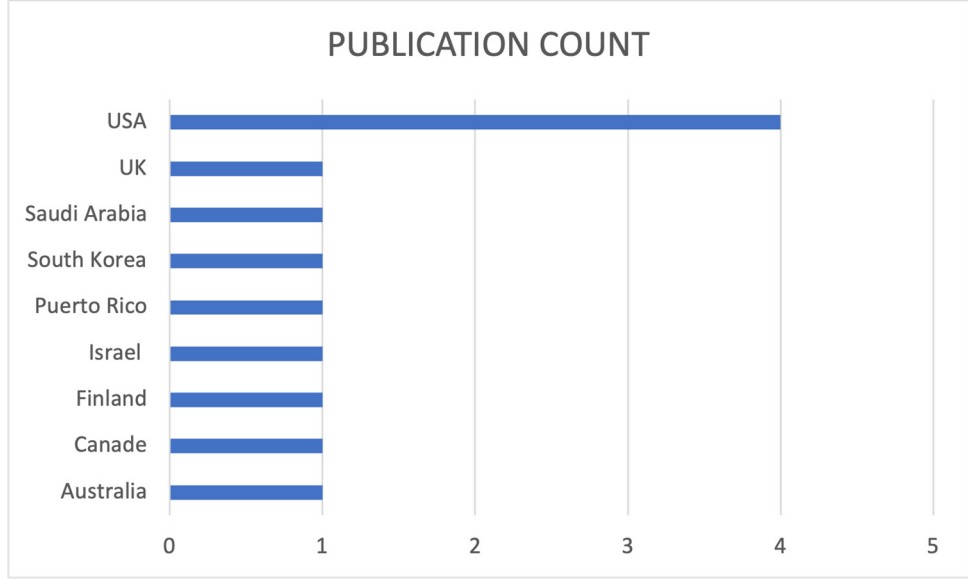

**Figure 2.** Studies categorized by country of origin.

With just 12 publications included in the review analysis, Figure 3 illustrates how few published articles there are overall. The number of publications fluctuates between different years. The number of publications in 2008 and 2021 is higher than other years. This fluctuation may reflect developments and changes in the research field. In addition,

the number of published articles has a certain distribution by year, indicating that research may receive attention at different times. This can result from the evolution of the field of study or topic.

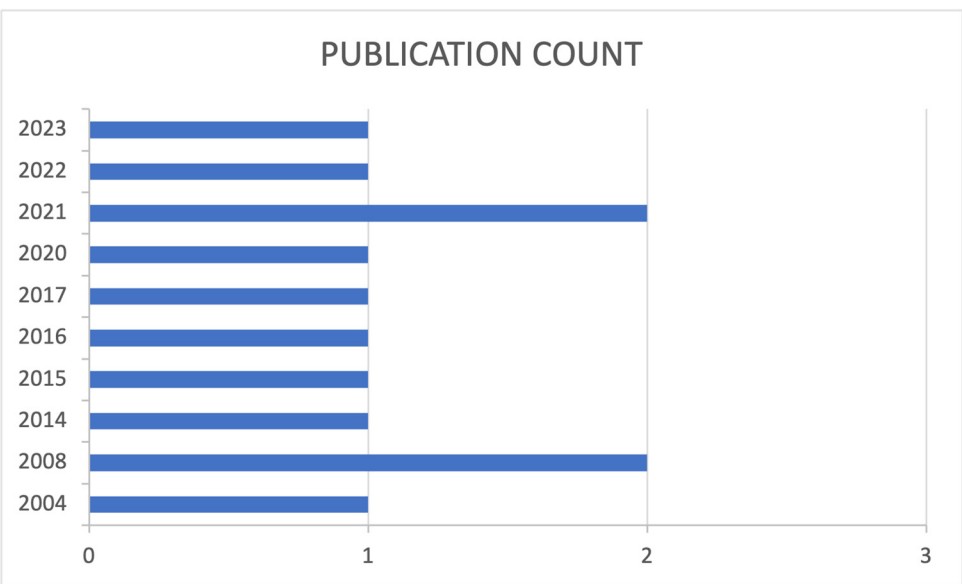

**Figure 3.** Publication trends over time: quantity and years.

Three factors may have contributed to this review's limited selection of articles. First, this study only considers the analysis of the effectiveness of different delivery modes of CPD, which is limited to specific research fields. Second, this study strictly limits the research objects, and the research subjects of the articles included in the review were all science teachers. Third, the research team restricted the included documents to those written and published in English.

*4.1. Comparative Effectiveness of Face-to-Face and Online CPD*

Empirical studies have shown that the online delivery mode for CPD training for science teachers is just as effective as the face-to-face training mode. In some specific situations, online CPD is more effective than traditional CPD [19]. Binmohsen and Abrahams [21] conducted an empirical study using mixed methods in the context of Saudi Arabia to evaluate the effectiveness of different delivery modes comparatively. To ensure that the training content was roughly the same, this study set up two control groups, A and B, for comparative research. This study is highly representative. Group A adopted the traditional delivery mode, while Group B adopted the online mode. After the participants had completed their training, they filled out a questionnaire survey and participated in interviews. The findings indicate that online CPD is as effective as learning in a face-to-face environment and that the online method effectively resolves participants' conflicts related to schoolwork schedules. It is more advantageous in the culture of Saudi Arabia, especially for female teachers. Many other studies have confirmed this finding; in other words, they show that the effectiveness of face-to-face and online CPD in cultivating teachers' quality and ability is consistent [46].

Ravitz et al. [35] identified seven common characteristics of effective professional learning for teachers in professional development programs: "(1) Is content focused; (2) Incorporates active learning utilizing adult learning theory; (3) Supports collaboration, typically in job-embedded contexts; (4) Uses models and modeling of effective practice; (5) Provides coaching and expert support; (6) Offers opportunities for feedback and reflection; (7) Is of sustained duration" [34]. Hammond and his team used conventional criteria for assessing CPD effectiveness to gauge the efficacy of online CPD and used a mixed-research approach to analyze the effectiveness of online CPD, with six of the seven characteristics fully sub-

stantiated. However, the sixth feature, which provides an opportunity for feedback and reflection, failed to receive adequate confirmation. Nevertheless, the research findings reflect on teaching design, with no inclusion of reflection on implementation.

### 4.2. Factors Influencing Different CPD Delivery Modes

#### 4.2.1. Factors Affecting Face-to-Face CPD Effectiveness

The first factor affecting the efficacy of the face-to-face CPD mode is the delivery cost, followed by the substance of the delivery.

In terms of delivery costs, traditional CPD entails comparatively high expenses. Teachers participating in the training often need to coordinate with the leaders of the institutions and the educational organization and modify the original teaching schedule before they can participate in CPD training [38]. Moreover, the traditional types of CPD often require participants to gather in designated areas and attend training sessions at specified times, directly leading to travel expenses generation [52]. Participants have to invest time and money to participate in face-to-face CPD.

In terms of content delivery, traditional CPD evaluations include many negative comments in terms of content; for example, it is noted that the training content is mostly theoretical, that the training content mostly revolves around standardized content, and that the training content lacks pertinence and is only weakly related to the profession of the trained teachers [38].

Above all, the impact of the effectiveness of face-to-face CPD can be summarized as follows: time costs, financial expenditure, the standardization and professional relevance of the training content, and support requirements for institutions and leadership.

#### 4.2.2. Factors Affecting the Effectiveness of Online CPD

It is worth noting that the jury is still out regarding the practical impact of online CPD on teachers and students. Previous research showed that the factors that may affect the effectiveness of online CPD are not related to the delivery mode but constitute other factors.

There are many influencing factors; the first factor that cannot be ignored is a stable internet connection. Participation in an online CPD requires a computer with stable access to the Internet, but not all users can afford this; in developing countries in particular, this is still a major factor affecting the effectiveness of online CPD [21,52].

Ravitz et al. [35] revealed the factors that may affect the effectiveness of online CPD, including the personal background of the teachers participating in the training, their different competencies and abilities, and the results of the interaction between attitudes and the participants' working environment. However, the study conducted by Ravitz et al. has certain shortcomings. For example, some socio-psychological variables were researched, but they ignored whether socio-demographic variables could potentially affect the effectiveness of online CPD. The research conducted by Mary and Cha [50] during the COVID-19 epidemic made up for this shortcoming. Using quantitative research methods, they pointed out that self-efficacy is a factor that can affect the effectiveness of online CPD and compared the effects of different genders and teaching experiences on self-efficacy. The study also found that online CPD containing UDL design elements, especially the webinar mode, positively impacts science teachers' self-efficacy, thereby promoting positive teaching practices. In addition, another factor that affects the effectiveness of online CPD is establishing a communication community. Because of the technical aspects of online continuing education, instructors must communicate with participants via the Internet. Creating a community of real-time communication between instructors and participants makes it challenging. It also hinders communication among the participants themselves, which, in turn, affects their ability to exchange ideas and learn from one another. As a result, they have difficulty in interactively gaining insights that enhance learning [46]. Communication can effectively promote positive values and attitudes in teaching and learning.

Another factor mentioned by only one author out of the 12 articles reviewed was the role of trainers in online CPD. In an online CPD program, the training teachers are different from the teachers in the face-to-face mode. Training teachers adjust the training method according to the type of participants involved in the training. These detailed changes are more likely to be based on personal guidance.

### 4.3. Advantages of Different CPD Delivery Modes

Regarding the advantages of the two different delivery types, the main advantage of face-to-face CPD over online CPD is that traditional CPD provides a communication community for participants. They can communicate face-to-face with trainers in real time, and problems can be solved in real-time. Participants can also engage in real-time communication regarding the training content and receive feedback through peer interaction, thereby enhancing their learning experience [49].

The various advantages of online CPD make it a common form of learning and career development. First, online CPD has no geographical restrictions, which means participants can easily access online CPD resources worldwide, thereby gaining a wider range of professional knowledge and experience. Second, it is cost-effective, reducing the cost of travel and printing materials. Third, flexible learning allows participants to arrange their professional learning time flexibly according to their working schedule. Based on these three points, participants can flexibly arrange the time and location of professional learning according to their schedules without commuting costs. In addition, online CPD participants can access specialized professional development training materials aligned with their respective research fields for personalized learning. This addresses the limitations associated with the broad, generalized content often found in traditional CPD training programs [53,54].

Although online CPD has also established an online communication community, the research indicates that community interactions in the online mode cannot be compared with the face-to-face communication community established in the traditional delivery mode. The traditional mode has more significant features in terms of community interaction.

### 5. Discussion

The main goal of this literature review was to provide a comprehensive analysis of the comparative effectiveness of different modes of delivering CPD for science teachers. Using the PRISMA method, the research team retrieved four databases and selected a total of 12 articles for analysis after conducting a quality assessment. This article employed two methods to analyze the effectiveness of face-to-face CPD and online CPD. The first section offers a quantitative descriptive analysis of the overall article, analyzing the country and year of publication. The second section encompasses a qualitative analysis with a primary focus on the following three dimensions: assessing the effectiveness of diverse delivery modes in CPD, examining factors influencing the efficacy of different CPD delivery modes, and a comparative evaluation of the merits associated with different delivery approaches.

This article offers a relatively comprehensive analysis of the effectiveness of different CPD delivery modes for science teachers. Both were equally effective regarding the effectiveness of CPD delivery types for science teachers. However, the factors influencing effectiveness are not inherent to the delivery mode but depend on external variables. Factors influencing the effectiveness of face-to-face CPD include working schedules, financial expenditures, institutional and leadership support, and a lack of specialized content standardization. Factors that affect the effectiveness of online CPD include a stable network infrastructure, participants' psychological variables, such as attitudes, beliefs, and self-efficacy, and the influence of the work environment and training teachers involved in online CPD programs.

Subsequently, the research team compared the advantages and differences between the two delivery modes. The primary benefit of in-person CPD is its capacity to foster an interactive learning community among participants. This mode enables direct, face-to-face

communication between participants and trainers, facilitating prompt problem-solving and engagement. Moreover, participants can engage in real-time communication and discussions to enhance their learning experience. This is difficult to achieve via online modes. However, online CPD also built a real-time virtual network community in which participants could communicate; this kind of online community often required participants to have digital competence, and the requirement for digital competence inevitably became a factor potentially affecting the effectiveness of online CPD. Compared with the traditional mode, the advantages of online CPD are much clearer. For instance, participants could overcome geographical restrictions and flexibly choose training times according to their working schedule; they could also choose more specific courses based on the relevance of their subjects and research fields. Online CPD offers cost-saving advantages over traditional delivery modes by eliminating expenses such as commuting costs.

This study comprehensively evaluates significant phases, integrating the research findings of two delivery modes, face-to-face CPD, and online CPD, and it systematically compares the impact of these two modes on science teachers. Additionally, the study synthesizes findings from the current research on factors influencing the effectiveness of different delivery modes. This research contributes to the general literature on teachers' continuing professional development and has implications for future research in CPD-related areas.

Although this study makes some contributions to the field of CPD, we must acknowledge that it has some limitations. First, the articles included in this study were retrieved from only four databases: Web of Science (WoS), Scopus, ERIC, and ScienceDirect. Additionally, the scope of the study was relatively narrow, and the choice of keywords was not inclusive enough, as it only focused on empirical research on the effectiveness of different delivery modes for science teachers. The selection criteria were restricted to research articles and conference papers, resulting in only 12 articles meeting the review criteria after a systematic screening process. This may have led to the exclusion of potentially valuable information. Furthermore, our review was confined to papers written and published in English, potentially overlooking valuable articles published in other languages.

Despite the limitations of this study, it suggests two main directions for future research: First, to examine the impact of social–psychological variables on the effectiveness of different CPD delivery modes; second, regarding the comparative analysis of online CPD and face-to-face CPD, it is crucial to underscore the profound impact of building a communication community on the effectiveness of CPD. This area constitutes a promising area for further investigation.

## 6. Conclusions

This article systematically reviews research findings regarding the effectiveness of diverse delivery modes for CPD for science teachers. It offers a comprehensive analysis and discussion encompassing three pivotal aspects: an evaluation of the effectiveness of various CPD delivery modes, an examination of the factors influencing the effectiveness of different CPD delivery modes, and a comparative assessment of the advantages associated with various delivery modes. A systematic review revealed that, when considering CPD delivery modes in isolation, the online and face-to-face methods have an equally effective impact on the professional development of science teachers. However, practical differences in their effectiveness are often attributed to external factors, including personal psychological variables and sociological factors. The sense of community created by the traditional CPD model plays a crucial role, and it is challenging to replicate it in online CPD. Additionally, digital competence may influence the effectiveness of online CPD.

## 7. Implications

External factors, such as personal psychological variables and sociological factors largely influence the variations in the effectiveness of CPD delivery modes. Recognizing the significance of these factors in practice is essential to enhancing the effectiveness of

various CPD modes. In addition, future research should focus on strategies to effectively enhance participants' communication and engagement in online CPD communities. Exploring the potential relationships between psychosocial variables, the effectiveness of CPD modes, and the impact of digital skills on online CPD effectiveness is an avenue for further investigation. While this article primarily focuses on science teachers, future studies can expand the scope of this research to include a broader range of subjects, facilitating more comprehensive discussions.

**Author Contributions:** Conceptualization, Z.L.; methodology, Z.L., N.C.H. and H.A.J.; software, Z.L.; validation, N.C.H. and H.A.J.; formal analysis, N.C.H. and H.A.J.; investigation, N.C.H. and H.A.J.; data curation, Z.L.; writing—original draft preparation, Z.L.; writing—review and editing, Z.L.; supervision, N.C.H. and H.A.J. All authors have read and agreed to the published version of the manuscript.

**Funding:** This research received no external funding.

**Institutional Review Board Statement:** Not applicable.

**Informed Consent Statement:** Not applicable.

**Data Availability Statement:** No new data were created or analyzed in this study. Data sharing is not applicable to this article.

**Acknowledgments:** I would like express my sincere gratitude to Seyedali Ahrari for his valuable insights during the writing of this paper. His expertise and thoughtful suggestions have significantly contributed to improvement of the manuscript.

**Conflicts of Interest:** The authors declare no conflict of interest.

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
