# Peer review of "The Effectiveness of Face-to-Face versus Online Delivery of Continuing Professional Development for Science Teachers: A Systematic Review"

_education, doi:10.3390/educsci13121251_

Round 1

Reviewer 1 Report

Comments and Suggestions for Authors

The study is well written and well presented. People could reproduce this study. There are some areas that need clarification.  But I think this is a competent study.

In the abstract it would be good to say 12 articles selected from a potential 82.  

Define what is exactly meant by in-person (does it include one-to-one, group learning, mentoring, coaching bespoke to individual, in-house, externally delivered) and online (humans meeting on zoom or Teams, self-directed training packages, blended learning)?

Lines 68-75 explain the reasons why science teachers were selected and while there is no issue with selecting one type of teacher it would be worth saying what does a science teacher include (engineering, mathematics, biology, medical, physics, craft and technology,  chemistry, sustainability?). Also, not everyone would agree with the statements that are generalisations. In some countries the creative industries are also very important to the economy. In the UK there is a shortage of all teachers including those working in the arts.  So, I think the reasons for doing the study should be softened a little. Also, are there pragmatic or biographical reasons for doing this study - do the researchers come from a science background?

Line 170 – states that relevant literature published up to October 12, 2023 was considered, but was there a start date for the range of published articles? Were articles considered from the last 20, 15 or 10 years?

Author Response

Response 1:
Thank you for pointing this out about the article’s abstract revising suggestion. I agree with this comment; hence I revise my abstract that is highlighted- Page 1, Line 12.

Response 2:
I have appended a concise paragraph to elucidate key terminologies within the introductory section of the article, aiming to enhance reader comprehension and facilitate a clearer understanding. - Page 3-4, Line 148-160.

Response 3:
We sincerely appreciate your valuable suggestions for refining the abstract. Following your guidance, we have implemented the necessary adjustments in the manuscript. It's worth noting that all authors involved in this paper have a solid scientific background. - Page 2, Line 72-75.

Response 4:
While conducting the literature search for this article, the start date for the search was not specified. The search deadline was set for relevant literature published as of October 12, 2023. However, it is important to note that the ERIC database only allows searches dating back to 2004, as illustrated in Table 3. - Page 6, Line 231.

Reviewer 2 Report

Comments and Suggestions for Authors

The authors are well done for exploring this topic.

The name and abstract include abbreviation (CDP), which is not very good for perception. The decoding goes only further in the section "keywords" and "introduction"

A very difficult topic. Especially with regard to measurements.

This article conducted a systematic review of research findings regarding the effectiveness of diverse delivery modes for CPD for science teachers. The authors explain the choice of teachers of this focus/profile. The authors clarify that this article primarily focuses on science teachers, future studies can expand the scope of this research to include a broader range subjects, facilitating more comprehensive discussions.

The algorithm of actions and the choice of articles is presented in sufficient detail, although I would like to increase the sample.

It seems to me that a little attention has been paid to the results themselves (in terms of volume). Some of the figures in the Results section can be assigned to the data chapter.

Author Response

Thank you very much for taking the time to review our manuscript. We sincerely appreciate the constructive feedback you provided, and we have made corresponding revisions to the paper. In order to provide a clearer expression of our responses, we have detailed the explanations for the modifications in the resubmitted files, with the changes highlighted for emphasis.

If you have any further suggestions regarding our responses or modifications, or if there are any concerns about our explanations, we would be more than willing to hear them and make further adjustments. We look forward to your continued guidance and hope that this round of revisions enhances the completeness of the manuscript.

Response 1:

Thank you for pointing out the mistakes in the title and abstract. As a result, I have made necessary adjustments to the title and abstract sections. - Page 1, Line 3- 6.

Response 2:

Thank you for your suggestions. May I try to keep the current results structure of the article? If it is not acceptable, please let me know, and I will make the necessary adjustments promptly.

Reviewer 3 Report

Comments and Suggestions for Authors

This is a systematic review focusing on the effectiveness of face-to-face versus online delivery of CPD for science teachers. The authors adopted the PRISMA methodology, however with several weaknesses. What I find more problematic for this review is that the choice of keywords is not inclusive enough and thus the results do not seem to cover a sufficient part of the relevant research. Specific comments are listed below.

23-142: I think you should discuss results of previous research in your introduction. Refer to previous relevant research on online CPD for science educators and discuss the results of previous research on factors that could potentially impact the efficacy of diverse forms of CPD programs and on the advantages of different CPD delivery modes. Moreover, discuss previous relevant reviews and argue on the novelty of your approach.

Table 1, inclusion criteria (1) and (2): Reading the inclusion criteria (1) and (2) I think they could be merged in order to be more clear. The way they are stated now, they may be misleading for the reader.

Table 1, inclusion criterion (3): If an article focuses on the effectiveness of in-person but not online CPD in improving teacher quality and practice is included or not? Please explain whether the articles should focus on the effectiveness of both in-person but not online CPD.

Table 1, inclusion criterion (4): Please, explain the difference between a science teacher and a science educator.

213-221: Explain how many researchers worked for the selection in each step, whether they worked independently and report on the agreement among them.

Figure 1: Please, explain which inclusion criteria were not met by the 23 records that were deleted before screening. Also, please explain which inclusion criteria were not met by the 29 records deleted for the next step.

Table 5: It is not clear whether the answer is YES or NO in each question for every record. It seems that there are different questions for different records and there are no questions at all for some of the records.

Figure 2. What is the need for 1.5, 2.5 etc in the x axis?

279-281: The COVID-19 pandemic does not seem to be a possible reason for the differences between years since during years 2015, 2016, 2017 (still no SARS-CoV-2 present) the amount of publications was the same as in the year 2020 (the peak of the pandemic).

297: Saudi Arabia is not listed in figure 2.

300-301: “Group A adopted the traditional delivery mode”. Here you use a different term to refer to the face-to-face delivery mode. I am noticing this to argue that using the criteria "Face-to-face" OR "Online" without other synonyms (such as traditional you use here) resulted in few results and I think you should have more in order for your discussion to be of value for relevant research.  

Comments on the Quality of English Language

Only minor, if any, editing of English language required.

Author Response

Response 1:

Thank you for sharing your insights on article writing ideas. Regarding the sections you mentioned, such as "Findings from previous research on factors that may influence the efficacy of various forms of CPD programs and the advantages of different CPD delivery models," in the ‘4.Results’ section of the article, there is a concentrated discussion. As you suggested. In light of your valuable comments, I have also incorporated additional remarks in the introduction section." I made some minor adjustments to improve the flow and clarity of the text. –Page2, Lines 58-61.

Response 2:

I have combined the inclusion criteria (1) and (2), as illustrated in Table 1.-Page 4, Lines 184-185.

Response 3:

Thank you for pointing out the issues. In order to reduce ambiguity, I have rephrased the statement regarding the inclusion criteria,as illustrated in Table 1.-Page 4, Lines 184-185.

Response 4:

Thank you for pointing out the issues. In order to reduce ambiguity, I have rephrased the statement regarding the inclusion criteria, as illustrated in Table 1.Page 4,Lines 184-185.

Response 5:

The difference between a teacher and an educator is primarily semantic. When compared with educator, the teacher typically refers to a job title; a teacher is a person who teaches in a school. But, an educator is a person who educates students. An educator is a person who provides instruction or education. An educator is usually seen as a mentor, instructor, or trainer. That is to say, an educator does not merely teach specific facts about an academic subject; an educator also instructs students' intellectual, moral, and social growth.

Response 6:

Thank you very much for pointing out the incompleteness in explaining the research process. I have now added a detailed description in the articles regarding the involvement of research team members in the selection process. I truly appreciate your suggestion, as it will contribute to enhancing the clarity and completeness of the document. - Page 4,Lines 227-230. 3

Response 7:

Thank you very much for pointing out the incompleteness in the article. I have provided a detailed description of the criteria used for excluding articles. - Page 6, Line 238-242.

Response 8:

According to the MMAT evaluation principles, the included articles should first be classified based on research methods. There are 5 evaluation questions for each category, see details in Table 5.
For example, (Binmohsen & Abrahams, 2020) and (Bitan-Friedlander et al., 2004) two studies belong to the category of Mixed-methods studies, and the evaluation questions are the same, so "—" is used in the table to refer to the same question, " ✓” indicates that the standard is met.

Response 9:

I have modified the problem in figure 2. – Page12, Line 299 .

Response 10:

I have removed controversial sentences and revised the sentence expressions. – Page12, Line 302 .

Response 11:

Thank you sincerely for your meticulous review of the article. I have carefully examined and addressed the affiliations, replacing the images, and refining the article descriptions to meet the standards. -Page11, Line293-297 .

Response 12:

Thank you for your valuable insights into the research of this article. Few studies exist on comparative groups analyses of face-to-face and online Continuing Professional Development (CPD) with control groups. The majority of existing studies are limited to empirical assessments of the effectiveness of single mode. The current literature review identified only one article for science teachers.

Round 2

Reviewer 3 Report

Comments and Suggestions for Authors

I appreciate the effort of the authors to address my concerns. However, there are still two major and one minor concerns unaddressed in my opinion. The minor point would be the difference between an educator and a teacher. The author's arguement is not supportive of their choice to distinguish between the two terms. I would suggest to keep just the one of the two words, e.g. educator. Shifting to the major points, Table 5 does not make sense to me. I assume that all the criteria of column "Methodological quality criteria" should be applied to all the papers but this is not shown in table 5. For instance, which criteria are applied to (Bitan-Friedlander et al., 2004) and (Stevenson et al., 2015)? I see the ticks under "YES" but the criteria cells are blank. Last but not least, I still think that the choice of keywords is not inclusive enough, as I argued in my previous review. I think that the authors should acknowledge that as a limitation at least.

Comments on the Quality of English Language

Just minor editing needed.

Author Response

Response 1

Thank you for your valuable suggestions. In addressing the issue concerning teachers and educators, I have made the necessary adjustments by retaining only "teachers" in the text. Additionally, I have modified the search string to ensure the overall integrity of the article. Please find these changes on Page 5, Lines 212-220.

Response 2:
Thank you for your suggestions. I have completed the questions about the standard cells of Table 5. We kindly request that you maintain Table 5, as both Table 4 and Table 5 are crucial stages for the quality assessment of the MMAT method. The inclusion of these tables reflects the comprehensive nature of the MMAT assessment method.

Response 3:
Thank you for pointing out that my keywords are not inclusive enough. I sincerely appreciate your suggestion, and to address this oversight, I have added a statement acknowledging this limitation at the end of the article. Page5, Line471-472.

Round 3

Reviewer 3 Report

Comments and Suggestions for Authors

The article may be published in the present form.

Comments on the Quality of English Language

Only minor editing may be needed.